# RETHINKING EFFECTIVENESS OF UNSUPERVISED DOMAIN ADAPTATION METHODS

## ABSTRACT

Recently, significant progress has been made in unsupervised domain adaptation (UDA) through techniques that enable reduction of the domain gap between labeled source domain data and unlabeled target domain data. In this work, we examine the diverse factors that may influence the effectiveness of UDA methods, and devise a comprehensive empirical study through the lens of backbone architectures, quantity of data and pre-training datasets to gain insights into the effectiveness of modern adaptation approaches on standard UDA benchmarks. Our findings reveal several non-trivial, yet valuable observations: (i) the benefits of adaptation methods decrease with advanced backbones, (ii) current methods under-utilize unlabeled data, and (iii) pre-training data matters for downstream adaptation in both supervised and self-supervised settings. To standardize evaluation across various UDA methods, we develop a novel PyTorch framework for domain adaptation and will release the framework, along with the trained models, publicly.

## 1 INTRODUCTION

Deep neural networks for image classification often suffer from dataset bias where accuracy significantly drops if the test-time data distribution does not match that of training, which often happens in real-world applications. To overcome the infeasibility of collecting labeled data from each application domain, a suite of methods have been recently proposed under the umbrella of unsupervised domain adaptation (UDA) (Hoffman et al., 2013; Long et al., 2015; 2017; Bousmalis et al., 2017; 2016; Long et al., 2018; Ganin & Lempitsky, 2015; Saito et al., 2017b; 2018; Zhang et al., 2019; Hoffman et al., 2018; Xu et al., 2019; Jin et al., 2020; Sharma et al., 2021; Kang et al., 2019; Wei et al., 2021; Kalluri et al., 2022; Berthelot et al., 2021) that allow training using only unlabeled data from the target domain of interest while leveraging supervision from a different source domain with abundant labels (Fig. 1a).

UDA methods have been greatly successful in improving the target accuracy on benchmark datasets under a variety of distribution shifts (Saenko et al., 2010; Peng et al., 2017; Venkateswara et al., 2017; Caputo et al., 2014; Peng et al., 2019). While literature in the area has focused on proposing new algorithms or loss functions, a holistic understanding of training choices that influence real-world effectiveness of domain adaptation has been lacking. In this paper, we make one of the first efforts to address this through an empirical study of three major factors that potentially influence performance the most, namely, 1. **Choice of backbone architecture:** With recent advances in architecture designs such as vision transformers (Dosovitskiy et al., 2020; Touvron et al., 2021; Liu et al., 2021) and improved CNNs (Liu et al., 2022b) we study which architectures suit domain transfer, and verify compatibility of existing adaptation methods with these backbones. 2. **Amount of unlabeled data:** Since the promise of unsupervised adaptation rests on its potential to leverage unlabeled target data, we study how much unlabeled data is really used by the adaptation methods. 3. **Choice of pre-training data:** We study whether pre-training the backbone on similar data as the downstream domain adaptation task is more beneficial than standard ImageNet pre-training, and examine this behavior across several supervised and self-supervised pre-training strategies.

Our study of a variety of UDA methods for image classification along those axes leads to several surprising observations:

- Recent advancements in vision transformers such as Swin (Liu et al., 2022a) and DeiT (Touvron et al., 2022b) exhibit superior robustness against diverse domain shifts when compared to the

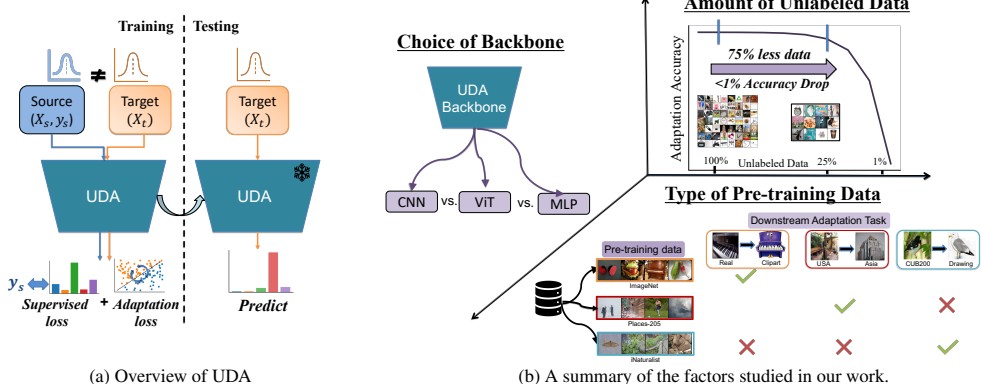

Figure 1: (a) **UDA overview:** Unsupervised domain adaptation allows training on unlabeled target data and labeled source domain using a combination of classification and adaptation losses to improve target test accuracy (Sec. 3). (b) **An overview of our empirical study:** We examine the effectiveness of SOTA UDA approaches through the lenses of backbone architectures (Sec. 5.1), unlabeled data quantity (Sec. 5.2) and nature of pre-training data (Sec. 5.3), and gather several useful observations.

conventional choice of ResNet-50 (see Tab. 1). When these advancements are incorporated into the backbone, the benefits provided by modern UDA methods tend to diminish, often resulting in significant changes to the relative ranking among the methods (see Fig. 3 and Sec. 5.1).

- For many UDA methods, reducing amount of target unlabeled data by 75% only resulted in $\sim 1\%$ drop in target accuracy (see Fig. 4), suggesting that existing approaches may not be adequate for enhancing performance, especially in scenarios where an increasing amount of inexpensive unlabeled data becomes accessible (see Sec. 5.2).

- Pre-training data matters for downstream adaptation, but in different ways for supervised and self-supervised pre-training. In supervised setting, pre-training on data similar to that used in the downstream domain adaptation task (*in-task data*) significantly improves the accuracy of both the baseline as well as all UDA methods compared to standard ImageNet pre-training (see Tab. 2), even when pre-training on the same amount of data.

- In self-supervised setting, pre-training on object-centric dataset enhances accuracy for object-centric adaptation tasks, while pre-training on scene-centric data is more suitable for scene-centric adaptation benchmarks (see Tab. 3). This trend holds across different types of pre-text tasks used in unsupervised pre-training (see Sec. 5.3).

We provide a summary of the findings from our empirical study in Fig. 1b. We believe such insights into the behavior of UDA methods have been previously hindered due to local choices of adaptation-independent factors like initialization, learning algorithm and batch sizes. To address this, we develop a new PyTorch framework that allows standardizing multiple UDA methods for those factors. Our framework will be made publicly available to continue improving our understanding of the effectiveness of UDA methods.

## 2 RELATED WORKS

**Unsupervised Domain Adaptation** A majority of works in unsupervised adaptation aim to minimize some notion of divergence between the source and target domains estimated using unlabeled samples (Ben-David et al., 2010; 2006). Prior works studied various divergence metrics like MMD distance (Long et al., 2015; 2017; Hsu et al., 2015; Yan et al., 2017; Baktashmotlagh et al., 2016; Pan et al., 2010; Long et al., 2013; Tzeng et al., 2014), higher-order correlations (Morerio et al., 2017; Sun & Saenko, 2016; Sun et al., 2016; Jin et al., 2020) or optimal-transport (Courty et al., 2017; Redko et al., 2019; Damodaran et al., 2018), but adversarial discriminative approaches (Ganin & Lempitsky, 2015; Tzeng et al., 2014; 2017; Xie et al., 2018; Long et al., 2018; Tzeng et al., 2015; Chen et al., 2019a; Saito et al., 2018) have been the most popular. To address the issue of noisy alignment with global domain discrimination (Kumar et al., 2018), recent works adopt category-level (Saito et al., 2017a; Pei et al., 2018; Kang et al., 2019; Na et al., 2021; Du et al., 2021; Wei et al., 2021;

Cui et al., 2020; Prabhu et al., 2021), instance-level (Sharma et al., 2021; Kalluri et al., 2022), consistency-based (Berthelot et al., 2021) or cross-attention (Xu et al., 2021) based techniques to improve adaptation. While the primary focus of most of these works is on algorithmic innovations to improve adaptation, our emphasis in this paper instead lies in identifying several key method-agnostic factors that impact performance of UDA methods, and conducting a comprehensive empirical study along these factors to uncover valuable insights that facilitate a better understanding of these methods. While domain adaptive semantic segmentation is also popular (Tsai et al., 2018; Hoffman et al., 2018; Vu et al., 2019; Kalluri & Chandraker, 2022; Lai et al., 2022), we restrict focus on adaptation methods for image-classification in our study.

**Comparative Studies**   Many recent works aim to enhance our understanding of the factors impacting the success of state-of-the-art models through extensive empirical analysis. Within computer vision, these works span the areas of semi-supervised learning (Oliver et al., 2018), metric learning (Musgrave et al., 2020; Roth et al., 2020), transfer learning (Mensink et al., 2021), domain generalization (Gulrajani & Lopez-Paz, 2020), optimization algorithms (Choi et al., 2019), few-shot learning (Chen et al., 2019b), contrastive learning (Cole et al., 2022), GANs (Lucic et al., 2018), fairness (Goyal et al., 2022) and self-supervised learning (Goyal et al., 2019; Newell & Deng, 2020). Our work follows suit, where we carefully devise an empirical study to revisit several standard training choices in unsupervised adaptation and reassess the competence of modern adaptation methods. The works closest to ours in domain adaptation are (Kim et al., 2022), which explores the impact of scalability of pre-training methods and (Musgrave et al., 2022), which assesses UDA methods through fair hyperparameter searches. Different from these, our focus extends to several other factors that impact adaptation such as quantity of unlabeled data and nature of pretext data used in pre-training.

## 3   Background: Unsupervised Domain Adaptation

The task of unsupervised domain adaptation (UDA) aims to improve performance on a certain target domain with only unlabeled samples by leveraging supervision from a different source domain with distribution shift. Specifically, we denote the source domain data using $D_s = \{X_s, y_s\}$ where $X, y$ denote the images and corresponding labels. Similarly, the unlabeled target domain is denoted using $D_t = \{X_t\}$. We assume that the source images are drawn from an underlying distribution $P_s$, and the target images are drawn from $P_t$, and domain shift arises due to discrepency between the source and target distributions. Of particular interest is the case of covariate shift (Ben-David et al., 2010), when $P_s \neq P_t$, although other forms of shift have also been studied in literature (Tan et al., 2020; Kalluri et al., 2023; Azizzadenesheli, 2022; Garg et al., 2020; Alabdulmohsin et al., 2023). The task of UDA is then to learn a predictive model using $\{X_s, X_t, y_s\}$ to improve performance on samples from the target domain $P_t$ (see Fig. 1a). A common choice for the feature encoder in UDA is a ResNet-50 (He et al., 2015) model pre-trained on ImageNet (Russakovsky et al., 2015), while a 2-layer MLP is generally used as the classifier on top the learnt features. While recent literature in UDA focuses on novel training algorithms and loss functions to improve transfer, this paper instead aims to study their effectiveness under several important but often overlooked axes of variations pertaining to backbone architectures, unlabeled data quantity and encoder pre-training strategies.

## 4   Analysis Setup

**Adaptation Methods**   We show results using a variety of UDA methods ranging from classical (DANN (Ganin & Lempitsky, 2015), CDAN (Long et al., 2018)), modern (MDD (Zhang et al., 2019), MCC (Jin et al., 2020), AdaMatch (Berthelot et al., 2021)) to state-of-the art (MemSAC (Kalluri et al., 2022), ToAlign (Wei et al., 2021), DALN (Chen et al., 2022)) approaches. The choice of methods in our comparative study is not aimed to be exhaustive of all the adaptation methods, but rather sampled to cover a wide variety of model families including adversarial (Ganin & Lempitsky, 2015; Long et al., 2018), non-adversarial (Zhang et al., 2019; Jin et al., 2020; Chen et al., 2022), consistency based (Kalluri et al., 2022; Berthelot et al., 2021) and alignment based methods (Wei et al., 2021). As shown recently (Kalluri et al., 2022), many other methods such as MCD (Saito et al., 2018), ILADA (Sharma et al., 2021), MSTN (Xie et al., 2018), AFN (Xu et al., 2019), CGDM (Du et al., 2021), RSDA (Gu et al., 2020) and BSP (Chen et al., 2019c) yield lesser accuracies even compared

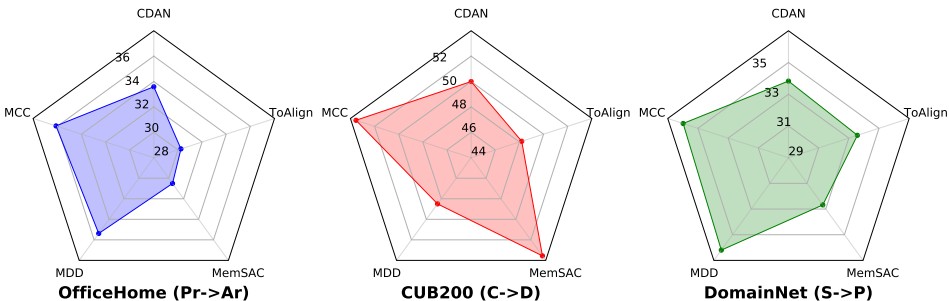

Figure 2: **Need for a joint framework** We illustrate the disparity between various codebases proposed for different UDA methods by noting their difference in accuracies for a plain source only model, which is computed without using any adaptation and should ideally match across implementations. We propose a new PyTorch framework for UDA to standardize evaluation across various methods and facilitate fair comparisons.

to a source-only baseline on many real-world datasets preventing their use in our study, although our inferences should readily transfer to any UDA method.

**Adaptation Datasets** We use visDA (Peng et al., 2017), OfficeHome (Venkateswara et al., 2017), DomainNet (Peng et al., 2019) and CUB200 (Wang et al., 2020) datasets in our analysis which are popularly used in UDA literature. visDA contains images for synthetic to real transfer from 12 categories, OfficeHome contains 65 categories across four domains, while DomainNet contains images from 345 categories from 6 domains. CUB200 is designed for fine-grained adaptation with images of 200 categories of birds from real and drawing domains. We additionally use the GeoPlaces dataset (Kalluri et al., 2023), which contains images from disparate geographies such as USA and Asia, in our analysis on effect of pre-training on downstream adaptation in Sec. 5.3.

**Evaluation Metrics** We report results using the accuracy on the test set of the target domain. In most prior works using OfficeHome and CUB200 datasets, the same set of data doubles up as the unlabeled target as well as the target test set used to report the results. To avoid possible over-fitting to target unlabeled data we create separate train and test sets for these datasets (using a 90%-10% ratio), and use images from train set as labeled or unlabeled data during training while reporting the final numbers on the unused test images. While this could lead slightly different numbers from those reported in the original papers, it also motivates fair comparison with the source-only baseline.

**Training Framework** We identify a problematic practice in most UDA methods where they are trained on different frameworks with different choices in various training hyper-parameters, making fair comparison across these works difficult. To highlight this issue, we compute the plain source-only accuracy using original code-bases of various UDA algorithms in Fig. 2 (the links to the open-source code for each of these methods are given in the supplementary). Essentially, we take the open-source code base for the methods, switch off all the adaptation losses, and train the model only on the source dataset to compute the target accuracy. Ideally, this accuracy, which acts as the baseline, should be the same across all the methods since it is independent of any adaptation. In practice, however, we observe that this baseline accuracy varies significantly between various UDA codebases, pointing to an underlying discrepency in various training choices adopted by these works unrelated to the adaptation algorithm itself. For example, unique to the respective methods, MDD (Zhang et al., 2019) uses a deeper MLP as a classifier, MCC (Jin et al., 2020) uses batchnorm layers in the bottleneck layer, CDAN (Long et al., 2018) uses 10-crop evaluation and AdaMatch (Berthelot et al., 2021) uses stronger augmentation on source data.

To alleviate this issue, we create a new framework in PyTorch (Paszke et al., 2017) for domain adaptation and implement several existing methods in this framework. Our framework standardizes different UDA methods with respect to adaptation-independent factors such as learning algorithm, network initialization and batch sizes while simultaneously allowing flexibility for incorporating algorithm-specific hyperparameters like loss coefficients and custom data loaders within a unified framework. All our comparisons and analyses in this paper are implemented using this framework, while using the same adaptation-specific hyperparameters proposed in the original papers in our re-implementation. Our framework will be publicly released to the research community to enable fair comparisons and fast prototyping of UDA methods in future works.

Table 1: Comparison of domain robustness of various vision architectures on standard adaptation datasets. We use the source accuracy ($\lambda_s$) and the target accuracy ($\lambda_t$) of a model trained only on source data to calculate the relative drop in accuracy ($\sigma_{st}=100*(\lambda_s-\lambda_t)/\lambda_s$, lower the better). Swin transformer shows consistently better robustness to domain shifts on several benchmarks.

| Model | ResNet-50 | Swin-V2-t | ConvNext-t | ResMLP-s | DeiT3-s |
|---|---|---|---|---|---|
| #Params | 24.12 M | 27.86 M | 28.10 M | 29.82 M | 21.86 M |
| DomainNet (R→C) | | | | | |
| Source Accuracy ($\lambda_s, \uparrow$) | 81.86 | 85.99 | 84.37 | 82.68 | 84.52 |
| Target Accuracy ($\lambda_t, \uparrow$) | 44.85 | 55.51 | 50.80 | 46.62 | 50.75 |
| Relative Drop ($\sigma_{st}, \downarrow$) | 45.21 | **35.45** | 39.78 | 43.61 | 39.95 |
| OfficeHome (Ar→Pr) | | | | | |
| Source Accuracy ($\lambda_s, \uparrow$) | 60.10 | 76.17 | 74.72 | 69.69 | 71.76 |
| Target Accuracy ($\lambda_t, \uparrow$) | 53.33 | 72.56 | 70.77 | 65.90 | 67.18 |
| Relative Drop ($\sigma_{st}, \downarrow$) | 11.26 | **4.74** | 5.29 | 5.44 | 6.38 |
| CUB200 (CUB→Draw) | | | | | |
| Source Accuracy ($\lambda_s, \uparrow$) | 81.00 | 87.75 | 85.88 | 84.62 | 88.12 |
| Target Accuracy ($\lambda_t, \uparrow$) | 52.60 | 58.90 | 52.74 | 53.41 | 56.36 |
| Relative Drop ($\sigma_{st}, \downarrow$) | 35.0 | **32.88** | 38.50 | 36.88 | 36.05 |
| GeoPlaces (USA→Asia) | | | | | |
| Source Accuracy ($\lambda_s, \uparrow$) | 57.17 | 63.11 | 60.39 | 58.99 | 61.65 |
| Target Accuracy ($\lambda_t, \uparrow$) | 36.12 | 42.53 | 40.30 | 38.11 | 40.34 |
| Relative Drop ($\sigma_{st}, \downarrow$) | 36.82 | **32.61** | 33.27 | 35.40 | 34.57 |

# 5 METHODOLOGY AND EVALUATION

## 5.1 WHICH BACKBONE ARCHITECTURES SUIT UDA BEST?

**Motivation**    While using ResNet-50 (He et al., 2015) is a widely adopted standard in many adaptation methods (Long et al., 2018; Saito et al., 2018; 2017a; Berthelot et al., 2021; Kalluri et al., 2022; Wei et al., 2021), we aim to study if recent advances in vision architectures such as vision transformers (Dosovitskiy et al., 2020) confer additional benefit to cross-domain transfer. While robustness properties of vision transformers to adversarial and out-of-context examples has been widely studied (Bai et al., 2021; Bhojanapalli et al., 2021; Shao et al., 2021; Zhao et al., 2021; Zhou et al., 2022), our analysis differs from these by focusing on the *cross-domain robustness* properties of these architectures instead on standard UDA benchmarks and investigate their potential as a substitute for the backbone in state-of-the-art adaptation methods.

**Experimental Setup**    Along with ResNet-50, we choose four different vision architectures which showed great success on standard ImageNet classification benchmarks: (i) DeiT (Touvron et al., 2021), (ii) Swin (Liu et al., 2021), (iii) ResMLP (Touvron et al., 2022a), and (iv) ConvNext (Liu et al., 2022b). We use newer versions of DeiT (DeiT-III (Touvron et al., 2021)) and Swin (Swin-V2 (Liu et al., 2021)) as they have better accuracy on ImageNet. We use the variants of these architectures which roughly have comparable number of parameters as ResNet-50, namely *DeiT-small*, *Swin-tiny*, *ResMLP-small* and *ConvNext-tiny*. All of them are pre-trained on ImageNet-1k, so their differences only arise from specific architectures. We use all pre-trained checkpoints from the timm library (Wightman, 2019) and architecture-specific training details are provided in the supplementary.

**Discussion**    For a model trained only on source-domain data (no adaptation), we use the accuracy on the source test-set ($\lambda_s$) and the accuracy on the target test-set ($\lambda_t$), to define relative cross-domain accuracy drop $\sigma_{st}=\frac{\lambda_s-\lambda_t}{\lambda_s}*100$. From Tab. 1, vision transformer architectures have the least value of $\sigma_{st}$ indicating better robustness properties compared to CNNs or MLPs. Specifically, *Swin-V2-t pre-trained on ImageNet-1k showed least relative drop ($\sigma_{st}$)* across all the datasets. Notably, on Real→Clipart from DomainNet, using Swin backbone with plain source-only training alone yields 55.5% accuracy, which is already higher than SOTA UDA methods that use ResNet-50 (54.5%) (Kalluri et al., 2022), indicating that *using a better backbone may have the same effect as using a complex adaptation algorithm* on the target accuracy.

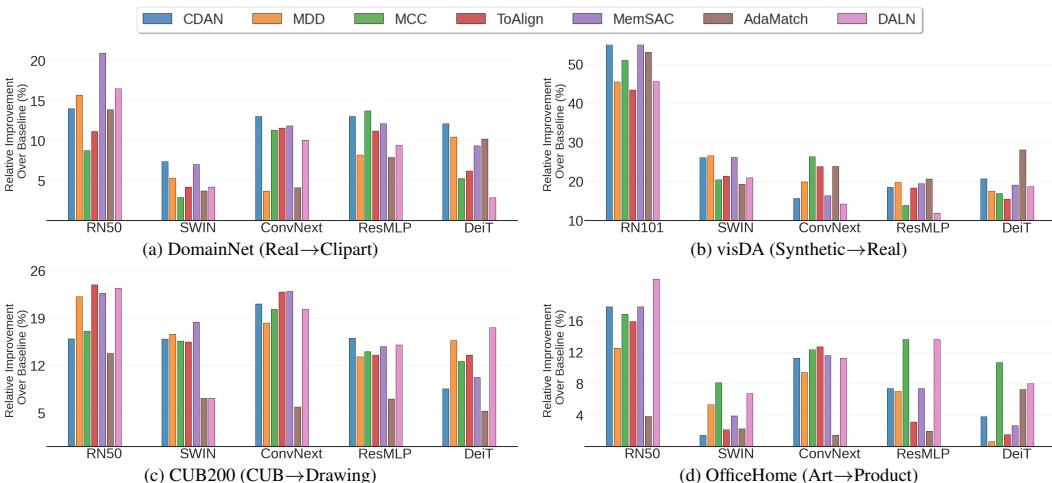

Figure 3: **Effect of backbone.** For each of the UDA methods, we show the gain in accuracy relative to a baseline trained only using source-data. Across datasets, we observe that the benefits offered by UDA approaches over the baseline diminish when using better backbones that have improved domain-robustness properties.

Next, we explore the viability of incorporating these advanced architectures into existing UDA methods. From Fig. 3, we observe that most methods do yield complimentary benefits over a source-only trained baseline even with newer architectures, but the *relative improvement offered by UDA methods over this baseline tends to diminish when using better backbones*. On Real→Clipart in Fig. 3a, the best adaptation method provides 20% relative gain over the baseline using ResNet-50, which falls to just 7% with Swin and 10% with DeiT backbone. Similarly, the relative gains offered by best UDA methods fall from 18% with ResNet-50 to 8% using Swin on Art→Product in Fig. 3d. These observation also holds for visDA Fig. 3b and CUB200 Fig. 3c datasets.

Furthermore, the *relative ranking of the best adaptation method changes across architectures and datasets*, and is not consistent. For example, an older and simpler method like CDAN gives best accuracies in Fig. 3a with Swin, ConvNext and DeiT, while MCC outperforms other methods with a ResMLP backbone. Likewise, MemSAC, CDAN and DALN yield best accuracy with Swin, ResMLP and DeiT backbones respectively on CUB200 in Fig. 3c, while MCC works best with most backbones on OfficeHome. Overall, these findings indicate that the effectiveness of many SOTA UDA methods is not entirely independent of the underlying backbone utilized, and often reduces in the presence of superior backbones that have stronger domain robustness properties.

## 5.2 How much unlabeled data does UDA need?

**Motivation** Although UDA algorithms hold great potential in leveraging unlabeled data from a target domain to enhance performance, an insight into their scalability properties in relation to the quantity of unlabeled data is lacking. This information is important to inform us which method has the greatest potential to improve performance when more unlabeled data becomes accessible. Therefore, we examine how much unlabeled data is actually necessary to train modern UDA approaches.

**Experimental Setup** To study the effects of data volume, we use $\{1, 5, 10, 25, 50, 100\}\%$ of the data from the target domain and run the adaptation algorithm using each of these subsets as the unlabeled data. For each fraction, we repeat the experiment with three different subsets and report the mean accuracy to eliminate sampling bias. Further, to avoid tail effects, we perform stratified sampling so that the label distribution is constant across all the subsets. Similar sampling strategies have been previously adopted in data volume ablation studies (Sohn et al., 2020). More details on our sampling procedure are provided in the supplementary. We note the possibility that the hyper-parameters in the respective methods are sensitive to the amount of target unlabeled data, but we do not preform any additional tuning to keep the number of experiments manageable. We restrict to using DomainNet and visDA in our analysis as those are the largest available datasets for domain adaptation. The already tiny data volume in OfficeHome and CUB200 prevents us from drawing any meaningful inferences appropriate for examining scalability suitable for our study. In supplementary, we show results on another recent large-scale adaptation benchmark GeoPlaces.

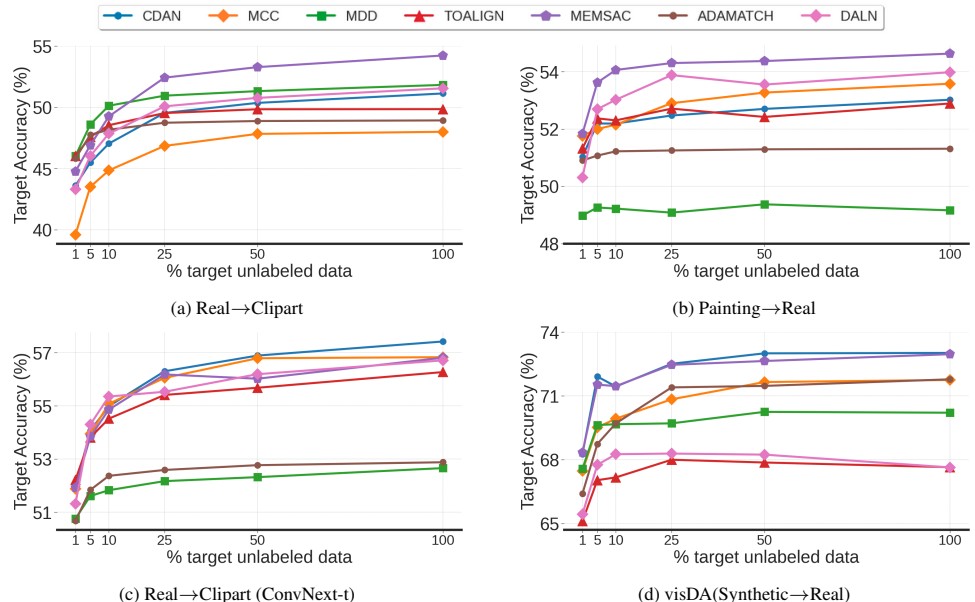

Figure 4: **How much unlabeled data do UDA methods need?** Across different adaptation datasets as well as different backbone architectures (Resnet50 in a, b, Resnet101 in d and ConvNext-tiny in c), we find that the performance saturates quickly with respect to amount of target data, showing their limited efficiency in utilizing the unlabeled samples. In most cases, even using 25% of the data only leads to less than 1% drop in accuracy.

**Discussion** Remarkably the trends from Fig. 4 indicate that on all the settings *the accuracy achieved by the unsupervised adaptation saturates rather quickly*. This trend holds for almost all of the studied adaptation methods, including adversarial and non-adversarial methods. For example, on R→C (Fig. 4a), the accuracy achieved at using just 25% of the unlabeled data is within 1% of the accuracy obtained at 100% of the data using any adaptation method. In P→R, (Fig. 4b) the accuracy plateaus much earlier, at around 10 − 15% of the unlabeled data. The improvements while adding additional unlabeled data stay less than 2% in most cases between 25% and 100% of unlabeled data. Using a different backbone like ConvNext (Fig. 4c) or different dataset like visDA (Fig. 4d) showed similar results, where the performance saturates after using only 25% of the unlabeled data.

We hypothesize that the main reason behind this observation is the underlying adaptation objective used in many works, which fails to effectively utilize growing amounts of unlabeled data. To verify this, we consider the objective of domain classification which forms the backbone of several adversarial UDA methods (Ganin & Lempitsky, 2015; Long et al., 2018), and examine its data efficiency.

We plot the accuracy of the domain discrimination objective against the quantity of unlabeled samples in Fig. 5 for different settings in DomainNet. We notice that the domain classification accuracy reaches a plateau after using approximately 25% of the data, potentially explaining the saturation of the adaptation accuracy in methods that rely on this objective for bridging the domain gap. We posit that this observation applies to other adaptation objectives as well, highlighting the need for novel objectives that can better utilize readily available unsupervised data to enhance adaptation. In the supplementary, we juxtapose this observation with a similar ablation using source labeled data, and identify that source supervision has a more pronounced effect on the final accuracy than target unlabeled data.

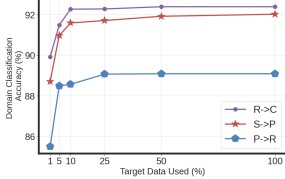

Figure 5: Domain classification accuracy vs. number of target samples (using ResNet-50).

## 5.3 DOES PRE-TRAINING DATA MATTER FOR UDA?

**Motivation** The availability of large pre-trained vision models has led to a widely adopted practice of utilizing these models for initialization and fine-tuning on downstream datasets of interest. Following recent works that revealed the importance of pre-training data in influencing downstream

Table 2: **Supervised pre-training and domain adaptation.** We analyze the relationship between nature of data used for supervised pre-training and the downstream adaptation task. Across all the methods, we observe that *in-task* supervised pre-training significantly helps adaptation. All models use ResNet-50 backbone.

| Pre-training | Plain Transfer (no adaptation) | | | ToAlign (Wei et al., 2021) | | | MemSAC (Kalluri et al., 2022) | | |
|---|---|---|---|---|---|---|---|---|---|
| | DNet | GeoP | CUB | DNet | GeoP | CUB | DNet | GeoP | CUB |
| ILSVRC-1M | **41.46** | 34.55 | 50.20 | **49.29** | 30.42 | 62.78 | **50.75** | 32.98 | 62.92 |
| Places205-1M | 35.14 | **41.95** | 40.83 | 38.55 | **34.9** | 55.29 | 41.93 | **40.16** | 54.22 |
| iNat2021-1M | 33.77 | 31.53 | **58.77** | 37.65 | 26.81 | **67.47** | 38.67 | 29.99 | **67.34** |

accuracy (Cole et al., 2022), we revisit the standard practice in UDA of employing ImageNet pre-training irrespective of the downstream adaptation task. This study contributes to revealing the properties that influence the selection of data and algorithms in various domain adaptation applications. While Kim et al. (Kim et al., 2022) share similar motivations as ours in their broad study of pre-training for adaptation, a notable distinction lies in their focus on *scaling* pre-training data and architectures. In contrast, our work provides complementary insights by exploring the relationship between the *type* of pre-training and downstream adaptation while maintaining a constant datasize.

**Experimental Setup** We use ImageNet (Russakovsky et al., 2015), iNaturalist-2021 (Van Horn et al., 2021) and Places-205 (Zhou et al., 2017) as datasets during pre-training. While ImageNet contains images from diverse natural and object categories, iNaturalist contains images of bird species and Places-205 is designed for scene classification. We select 1M images from each of these datasets (indicated as ImageNet-1M, iNat-1M and Places205-1M) to keep the size of the pre-training datasets constant, allowing us to decouple the impact of nature of data from the volume of the dataset. In terms of pre-training methods, we use supervised pre-training using labeled data, along with recent state-of-the-art self-supervised methods SwAV (Caron et al., 2020), MoCo-V3 (Chen et al., 2021) and MAE (He et al., 2022), which broadly cover the three families of clustering, contrastive and patch-based self-supervised learning. We train SwAV on ResNet-50, MoCo on ViT-S/16 and MAE on ViT-B/16 architectures, along with supervised pre-training on ResNet-50. More details on the training strategies followed for each method is provided in our supplementary material.

For the downstream adaptation tasks, we use Real→Clipart on DomainNet, CUB→Drawing on CUB200 and USA→Asia on GeoPlaces covering three distinct application scenarios for adaptation on objects, birds and scenes respectively. To prevent overlap between pre-training and adaptation data, we remove images from Places-205 that are also present in GeoPlaces and remove images from iNaturalist that belong to the same class as those in CUB200.

**Supervised Pre-training** In our analysis, we loosely consider pre-training on ImageNet, iNaturalist and Places205 to be *in-task pre-training* for downstream adaptation on DomainNet, CUB200 and GeoPlaces respectively. We show our results using supervised pre-training on Resnet-50 in Tab. 2 for three settings - plain source-only transfer (no adaptation), and adaptation using ToAlign and MemSAC. Across the board, we observe that *in-task pre-training always yields better results on downstream adaptation* even when using the same amount of data. Focusing on plain transfer from Tab. 2, the de-facto practice of using supervised ImageNet pre-training gives 50.2% on CUB→Drawing transfer task, while just changing the pre-training dataset to iNaturalist2021 yields 8.5% absolute improvement on target accuracy. Likewise, we observe a non-trivial improvement of 7.4% absolute accuracy for GeoPlaces by using Places205 for pre-training even without any adaptation, challenging the common assumption of using an ImageNet-pretrained model irrespective of the downstream task. It is notable that most UDA algorithms actually yield negative returns when applied on GeoPlaces. We hypothesize that supervised pre-training on in-task data creates strong priors and learns more relevant features, thereby enhancing generalization on similar downstream tasks. Consequently, we conclude that selecting in-task pre-trained models is a viable approach to improve transfer, particularly when target unlabeled data is unavailable. Our inferences follow similar recommendations recently made in other problems like robotics (Xiao et al., 2022) and medical imaging (Taher et al., 2021).

We also observe that the benefits obtained from in-task supervised pre-training complement the advantages potentially obtained using UDA methods, resulting in additional improvements in accuracy. From Tab. 2, on CUB200, we observe 17.1% and 17.3% improvement using MemSAC and ToAlign respectively together with in-task pre-training, over standard practice of ImageNet-pretraining and fine-tuning on source data (12% from changing the backbone and further 5% from the adaptation).

Table 3: **Self-supervised pre-training and domain adaptation.** We find that self-supervised pre-training on object-centric images (on ImageNet) help downstream accuracy on object-centric adaptation (on DomainNet and CUB200), while scene-centric pre-training (on Places205) benefit adaptation on scene-centric GeoPlaces task.

(a) **Plain Transfer (No Adaptation)**

| | SwAV (ResNet50) Caron et al. (2020) | | | MoCo-V3 (ViT-s/16) Chen et al. (2021) | | | MAE (ViT-b/16) He et al. (2022) | | |
|---|---|---|---|---|---|---|---|---|---|
| Pretraining | DNet | GeoP | CUB | DNet | GeoP | CUB | DNet | GeoP | CUB |
| ILSVRC-1M | **36.51** | 35.76 | **31.59** | **30.48** | 31.13 | **40.7** | **38.58** | 35.85 | **52.34** |
| Places205-1M | 30.86 | **42.26** | 27.44 | 27.45 | **35.89** | 39.49 | 34.76 | **38.1** | 45.25 |
| iNat2021-1M | 28.01 | 29.01 | 30.12 | 25.66 | 27.82 | 40.03 | 33.78 | 31.68 | 49.4 |

(b) **Using MemSAC Adaptation**

| | SwAV (ResNet50) Caron et al. (2020) | | | MoCo-V3 (ViT-s/16) Chen et al. (2021) | | | MAE (ViT-b/16) He et al. (2022) | | |
|---|---|---|---|---|---|---|---|---|---|
| Pretraining | DNet | GeoP | CUB | DNet | GeoP | CUB | DNet | GeoP | CUB |
| ILSVRC-1M | **44.6** | 36.33 | **51.81** | **34.33** | 30.35 | **52.61** | **44.91** | 34.07 | **64.26** |
| Places205-1M | 36.48 | **41.14** | 39.49 | 30.83 | **35.51** | 46.99 | 39.56 | **37.00** | 53.68 |
| iNat2021-1M | 31.6 | 28.75 | 45.65 | 28.24 | 26.01 | 48.46 | 38.48 | 28.74 | 59.7 |

On the other hand, a significant mismatch between the pre-training dataset and the downstream domain adaptation dataset (such as Places and Birds datasets), noticeably reduces the accuracy by $>10\%$ in most cases, underlining the dependence of model's generalizability to the pre-training data.

**Self-supervised Pre-training**   We show results for self-supervised setting in Tab. 3. We first note that supervised pre-training achieves much higher accuracies after downstream adaptation compared to self-supervised pre-training. This is expected, as supervised pre-training captures richer object semantics through labels inherently benefiting any downstream task, whereas self-supervised learning relies on pretext tasks that may not provide the same level of semantic understanding.

In terms of pre-training data, we observe that both CUB200 and DomainNet benefit from self-supervised pre-training on ImageNet, while GeoPlaces still benefits from pre-training on Places205. This observation holds for both source-only transfer (Tab. 3a) as well as adaptation using MemSAC (Tab. 3b). We posit that in a self-supervised setting, the nature of images in the datasets (whether object-centric or scene-centric) plays a crucial role in downstream transfer. Specifically, pre-training on object-centric images from ImageNet leads to improved image classification accuracies on DomainNet and CUB200. Conversely, unsupervised pre-training on scene-centric Places205 showcase better transfer performance in place recognition tasks on the GeoPlaces dataset. Among object-centric datasets, we find that the diversity of images in ImageNet is better for effective transfer compared to specific domain-based datasets like iNaturalist. The diversity in images from ImageNet is also shown to benefit self-supervised learning in general (Cole et al., 2022), potentially explaining away our observations. This property is consistent across different kinds of self-supervised pretext tasks like SwAV, MoCo and MAE. We emphasize that, in contrast to the findings of Kim et al. (2022) who revealed improvements using larger or multimodal datasets, we maintain a consistent scale across all datasets uncovering complementary characteristics pertaining to the *type* of the employed pre-training data. We focus on image-only pre-training here, and studying effects of vision-language foundation models (Radford et al., 2021) on domain adaptation is left to a future work.

## 6 CONCLUSION

In this paper, we provide a holistic understanding of factors that impact the effectiveness of image-classification based UDA methods, most of which are not apparent from standard training and evaluation practices such as ImageNet-pretrained ResNet backbones. We perform a comprehensive empirical study through the lens of backbone architectures, quantity of unlabeled target data and pre-training datasets on standard benchmarks like DomainNet, OfficeHome and CUB200, highlighting several key insights regarding the sensitivity of these methods to the backbone architecture, their limited efficiency in utilizing unlabeled data, and the potential for enhancing performance by opting for in-task pre-training. These observations shed new light into several areas where future research might be needed in order to improve adaptation. In terms of limitations, we only consider UDA designed for classification in this work, and our findings might or might not hold for other problem areas such as domain adaptive semantic segmentation. Likewise, conducting similar study into other types of transfer like universal adaptation (You et al., 2019), source-free adaptation (Kundu et al., 2020) and semi-supervised adaptation (Saito et al., 2019) is left to a future work. We also acknowledge the potential existence of other unexplored factors that may impact the performance of UDA methods, beyond those studied here.

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
