# OpenReview forum: "Rethinking Effectiveness of Unsupervised Domain Adaptation Methods"
_ICLR.cc/2024/Conference — ICLR 2024 Conference Withdrawn Submission_

### Official Review · Reviewer_iPUs · 2023-10-23

**Soundness:** 2 fair
**Presentation:** 2 fair
**Contribution:** 1 poor
**Rating:** 3
**Confidence:** 5

**Summary:**

The paper conducts a comprehensive empirical examination of seven unsupervised domain adaptation (UDA) methods for image classification across four popular UDA benchmarks. The study delves into three specific facets: the impact of backbone architectures, the influence of data quantity, and the effects of pre-training datasets. This investigation yields three key findings: (i) the advantages of modern UDA techniques decrease as backbone architectures advance, (ii) current UDA methods do not fully exploit unlabeled target data, and (iii) in-task pre-training boosts performance in downstream target domain adaptation.

**Strengths:**

**(i)** The paper is well-organized and easy to follow. For example, the methodology section neatly delineates the three primary factors the paper addresses.

**(ii)** The paper conducts numerous experiments, covering a wide range of domain adaptation benchmarks. This includes both classic benchmarks such as OfficeHome and visDA, and recent benchmarks like DomainNet and GeoPlaces.

**(iii)** The paper provides comprehensive details, including intricate model configurations, dataset specifications used in the experiments, and code to significantly enhance reproducibility.

**Weaknesses:**

**(i)** The paper lacks novel insights and original contributions. The majority of the "surprising observations" summarized have previously been introduced in existing publications. Specifically, the first finding, regarding how advanced backbones like Swin and ConvNext can influence the UDA method ranking with ResNet-50, was previously introduced in [1]. The second finding related to the leverage of unlabeled target data is not sound and is discussed in the second weakness below. The last two findings concern the effect of pre-training datasets in domain adaptation; however, [2] has comprehensively explored both supervised and self-supervised in-task pre-training in transfer learning and its specific application in domain adaptation. Additionally, [3] has provided a comprehensive sanity check on the effectiveness of UDA with 49 published UDA methods, which is far more comprehensive than the 7 UDA methods used in this paper. In the last part of the related work section, the authors mentioned [1, 3] and acknowledged that only two of their contributions are novel, namely, the study of the quantity of unlabeled data and the nature of pretext data used in pre-training. However, considering the missing related work [2], it appears that only the study of the quantity of unlabeled data is original, which significantly diminishes the paper's novelty.

**(ii)** The paper's summarized findings lack soundness because the experiments are not comprehensive and may present a one-sided view. First, the authors base their conclusions on only seven closed-set UDA methods in their empirical study while attempting to draw conclusions about the entire UDA field. In Section 4, the authors categorize all UDA methods into four types, which is an incomplete categorization compared to the more comprehensive studies found in [3, 4] in the field of UDA. In particular, the second observation, which claims that "reducing the amount of target unlabeled data by 75% results in only about a 1% drop in target accuracy," lacks soundness. This is because the paper overlooks relevant types such as pseudo-labeling/self-training UDA methods and input-space alignment methods [5], which are essential for a well-rounded analysis. Second, as suggested in [3], hyperparameter tuning plays a significant role in ensuring UDA methods perform well, and optimal hyperparameters often vary across different datasets and backbones. The paper utilizes the same hyperparameters, tuned with ResNet and common UDA benchmarks, for new backbones and datasets, making the comparison among UDA methods unfair and the observation less sound. Third, using normalized percent values of performance drop may not be convincing. This is because, given the same absolute accuracy drop, a large denominator in the percent calculation is likely to lead to a small percent value.

**(iii)** The paper's significance may be limited for a few reasons. Firstly, it provides empirical comparisons among selected UDA methods without offering a theoretical analysis of key UDA problems, such as covariate shift and the utilization of target unlabeled data. Consequently, it's not evident how this study contributes to the development of novel and advanced UDA methods. Second, the paper exclusively concentrates on closed-set UDA for image classification. While this is a valuable focus, it leaves unexplored the broader challenges in UDA, including scenarios with category shifts and UDA without source data. A more valuable contribution would involve an analysis of the pre-training dataset for UDA without source data, given that the pre-trained model is the sole source of information for the target domain.

**Reference**

[1] A Broad Study of Pre-training for Domain Generalization and Adaptation. ECCV 2022

[2] Self-Supervised Pretraining Improves Self-Supervised Pretraining. WACV 2022

[3] Unsupervised Domain Adaptation: A Reality Check. arXiv 2021

[4] A Survey of Unsupervised Domain Adaptation for Visual Recognition. arXiv 2021

[5] Cycada: Cycle-consistent Adversarial Domain Adaptation. ICML 2018

**Questions:**

Please refer to the weaknesses part for detailed questions and suggestions.

---

### Official Review · Reviewer_bJYV · 2023-10-26

**Soundness:** 2 fair
**Presentation:** 3 good
**Contribution:** 2 fair
**Rating:** 3
**Confidence:** 5

**Summary:**

This paper tends to explore some factors that may be related to the adaptation performance of unsupervised domain adaptation, rather than propose a special model in numerous UDA algorithms. This paper focuses on three factors: backbones, unlabeled target data and pre-training data, and observe some points. 1) Stronger backbones may weaken the adaptation benefits. 2) The performance stays stable when a certain unlabeled target data is used. 3) The performance depends on the relation between Pre-training data and down-stream tasks.

**Strengths:**

1. Among countless UDA models in recent years, it is nice to have a cool exploration what matters in UDA. It is clear that in numerous UDA models, the comparisons are not always fair which involves backbones (layers, hyperparemeters, optimizers, etc.), pre-training data, etc. as this submission mentions. Therefore, I acknowledge the effort made in this paper to reveal these issues.
2. Additionally, it is nice to observe that the unlabeled data does not make significant contribution to UDA models, which may inspire researchers explore how to fully utilize target data for improving adaptation performance. But I hold another different opinion about this observation, described in Weakness.

**Weaknesses:**

1. These factors explored in this submission and the observations seem common or general recognition by researchers. Actually, this is not special in UDA, and also holds in other computer vision branches. Backbones and pre-training data are certainly key factors for improving performance. Large models in recent months have well confirmed this opinion. Therefore, the essential contribution of this submission may be insignificant, although I acknowledge the effort made by authors.
2. Although it is interesting to find that the unlabeled target data actually does not make full effort and ignored by researches, it may not mean the unabeled data amount is not important in UDA. My opinion is related to the sampling strategy. If the authors adopts random sampling about 20% from the unlabeled target domain, it may not equal to that only 20% target data is seen. Actually, if the whole unlabeled target domain is seen, although the training target data amount becomes 20% the whole data, the target distribution is still observed. This may cause the performance stays stable when 50% target data is used for adaptation training.

**Questions:**

1. Since three factors have been studied, I would like to see what specifc suggestions and advices can be provided for unsupervised domain adaptation?
2. I also concern about the correctness of the possible suggestion. For example, larger backbones or stronger backbones can easily improve UDA performance. But from the perspective of large models, such as SAM, it does not mean it can well adapt to some downstream tasks. In other words, the performance may be weaker than small model on downstream tasks.

---

### Official Review · Reviewer_txAu · 2023-10-29

**Soundness:** 3 good
**Presentation:** 2 fair
**Contribution:** 3 good
**Rating:** 6
**Confidence:** 4

**Summary:**

The paper investigates the efficacy of current unsupervised domain adaptation (UDA) algorithms in the context of three changing factors: 1) model architecture, 2) volume of unlabeled data in the target domain, and 3) the selection of pre-training data. The authors present a comprehensive experimental analysis that compares seven distinct UDA approaches, encompassing diverse categories including adversarial methods, alignment-based methods, and consistency-based methods. These evaluations are performed across four benchmark datasets: VisDA, DomainNet, CUB200, and OfficeHome. The experiments reveal interesting results: 1) with vision transformer models the benefits of UDA methods diminish, 2) current UDA methods underutilize the target domain data, and 3) pre-training plays a crucial role in downstream adaptation performance. These findings are interesting and would be very useful to the UDA community. Additionally, to further encourage research in this direction, the paper also presents a new unified PyTorch codebase that serves to standardize different UDA methods and their respective hyperparameters, architectures etc.

**Strengths:**

1. With a multitude of UDA algorithms out there, it is difficult to choose or compare between these approaches. This is further compounded by the lack of standardized baselines and hyperparameter choices. This paper takes an important step in resolving these issues.

2. The paper also proposes several changes to the traditional practices followed in the literature: 1) shifting to vision transformer models such Swin from ResNet-50, and 2) using proper pre-training strategy based on the task at hand, instead of simply using ImageNet. I believe the research community will benefit from these insights.

**Weaknesses:**

1. Some of the presentation of the results can be improved. For example, it is not clear which algorithm achieves the absolute highest performance after UDA from the relative improvement results. We know that on the Swin architecture, MemSAC achieves highest relative improvement over the unadapted model, while on the ConvNext model CDAN achieves the best. But we do not know which one out of these two performs the absolute best.

2. The experiments are performed only on a subset of the domains of the benchmarks. The results would be stronger when averaged across all the domain combinations as done in the UDA literature.

**Questions:**

1. The standardized codebase seems to be missing some of the methods and dataloaders. Since this is a significant portion of the contribution, I would appreciate a full version being added to supplementary instead of a basic version. Can you please add this?

---

### Official Review · Reviewer_ew3y · 2023-11-10

**Soundness:** 3 good
**Presentation:** 3 good
**Contribution:** 2 fair
**Rating:** 5
**Confidence:** 3

**Summary:**

The authors investigate the influence of multiple factors on the effectiveness of UDN methods, including backbone architectures, scale of unlabeled target data, and pre-training data.

**Strengths:**

1.	The paper is straightforward and easy to follow.
2.	A new framework in PyTorch is created for the study of domain adaptation.
3.	Different baseline architectures are examined, including ResNet-50, DeiT, Swin, ResMLP and ConvNext.  The results show that the benefits of of UDN are reduced when with stronger backbones.
4.	The influence of scale unlabeled target data is examined and concludes that exiting UDA methods fail to fully leverage the large-scale unlabeled target data.
5.	The influences of supervised pre-training and self-supervised pre-training are investigated.

**Weaknesses:**

1.	The title may be changed to: “rethinking effectiveness of VISUAL unsupervised domain adaptation methods”, as only visual classification tasks are investigated.
2.	Limited inspirations are derived from the experimental analysis. For those who are familiar with UDN, the findings in this paper seem to be trivial and already well known. It’s expected that more deep analysis to be conducted to provide deep insights.

**Questions:**

None